# Quad-Mag Board for CubeSat Applications

Brady P. Strabel[1], Leonardo H. Regoli[1,2], Mark B. Moldwin[1], Lauro V. Ojeda[3], Yining Shi[1], Jacob D. Thoma[1,4], Isaac S. Narrett[1,5,], Bret Bronner[6], and Matthew Pellioni[1,7]

[1]Climate and Space Sciences and Engineering, College of Engineering, University of Michigan, Ann Arbor, MI, USA
[2]The Johns Hopkins University Applied Physics Laboratory, Laurel, MD, USA
[3]Mechanical Engineering, College of Engineering, University of Michigan, Ann Arbor, MI, USA
[4]Jet Propulsion Laboratory, California Institute of Technology, Pasadena, CA, USA
[5]Earth, Atmospheric, and Planetary Sciences, Massachusetts Institute of Technology, Cambridge, MA, USA
[6]Made In Space Incorporated, Moffett Field, California, USA
[7]General Dynamics Land Systems, Sterling Heights, MI, USA

**Correspondence:** Brady P. Strabel (bstrabel@umich.edu)

**Abstract.** The design, characteristics, and performance of a CubeSat magnetometer board (Quad-Mag) equipped with four PNI RM3100 magnetometers is presented. The low size, weight, power, and cost of the RM3100 enables the inclusion of four sensors on a single board, allowing a potential factor of two reduction in the noise floor established for an individual sensor via oversampling with multiple sensors. The instrument experimentally achieved a noise floor of 5.34 $nT$ (individual axis), averaging across each axis of the four magnetometers, at a 65 $Hz$ sampling rate. This approaches the theoretically established limit for the system of 4.37 $nT$ at 40 $Hz$. A single on-board, Texas Instrument MSP430 microcontroller handles synchronization of the magnetometers and facilitates data collection through a simple UART-based command interface to a host system. The Quad-Mag system has a mass of 59.05 $g$ and total power consumption of 23 $mW$ while sampling and 14 $mW$ while idle. The Quad-Mag enables nearly 1 $nT$ magnetic field measurements at 1 $Hz$ using commercial-off-the-shelf sensors for space applications under optimal conditions.

## 1 Introduction

Measuring magnetic fields in space is required to understand most heliophysics and space physics systems. From the interplanetary medium down to the upper layers of planetary ionospheres, the interaction between charged particles and magnetic fields defines the motion of charged particles, the convection of plasmas as well as the generation and damping of waves (Baumjohann and Treumann, 2012). An important limitation of traditional space missions when studying the dynamic nature of the space environment is the inability to sample more than one point in space at any given time. This makes it impossible to disentangle multiple signals from different sources.

In recent years, multi-spacecraft missions have been launched to study different aspects of the Earth's magnetosphere (e.g., Burch et al., 2016; Fear et al., 2014; Bandyopadhyay et al., 2015; Maruca et al., 2021; Friis-Christensen et al., 2006). Given the prominent role of magnetic fields, all of these missions were equipped with high resolution science magnetometers. Different

technology developments have led to smaller magnetometers with the capability of measuring fields with very high resolution, impossible to achieve a couple of decades ago.

The relative low costs associated with CubeSats makes them the natural choice for future multi-spacecraft studies (NASEM, 2016). However, due to their small size, any system designed to be used in a CubeSat needs not only to be small, but also to
have very low power consumption (due to the limited area for solar panels). In addition, in order for a CubeSat mission to take advantage of the low-cost concept, the production price for any instrument needs to be low.

A number of different approaches have been taken in order to obtain magnetic field measurements with a resolution sufficiently high to perform scientific studies of the magnetosphere. In general, these efforts can be summarized in two main categories, namely, miniaturization of traditional fluxgate and helium magnetometers (e.g., Miles et al., 2016; Guo et al., 2017;
Forslund et al., 2007) and the use of commercial-off-the shelf (COTS) sensors (e.g., Matandirotya et al., 2013; Brown et al., 2012, 2014; Novotny et al., 2021). The system presented in this paper takes the latter approach and describes not only the use of a low-size, weight, and power plus cost (SWAP+C) chip-based COTS magnetometer, but the combination of multiple magnetometers into a single system to improve the resolution by oversampling with multiple sensors. This expands on previous work, outlined in Regoli et al. (2018a), that was constrained by circuit board errors and therefore incomplete characterization.
Magnetic interference testing for the complete system is also introduced as another performance gauge.

## 2  Magneto-Inductive Sensing

The RM3100 magnetometer, manufactured by PNI Sensor Corporation, consists of three magneto-inductive (MI) sensors and a single control application-specific integrated circuit (ASIC). The MI sensors are a simple solenoidal coil wrapped around a highly permeable magnetic core. Incorporating the sensor with the control ASIC creates the basic resistor-inductor (RL)
sensing circuit (Figure 1) that drives the MI technology.

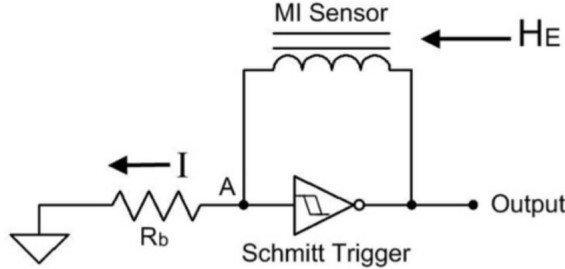

**Figure 1.** Schematic of the MI Sensor (from Leuzinger and Taylor (2010))

Magneto-inductive sensing hinges on the fact that the induction of a coil wrapped around a highly permeable magnetic core will fluctuate with respect to the magnetic field being applied to the coil. The magnetic field experienced by the coil (H), in

turn, consists of the external field parallel to the coil ($H_E$) and the field generated by current running through the circuit itself (I). It can be represented by the equation $H = kI + H_E$, where $k$ represents the conversion factor of the coil. With this in mind, the inductance of the sensor is clearly a function of the magnetic field, as seen in Figure 2.

The circuit in Figure 1 employs a Schmitt trigger with a bias resistor ($R_b$) and the MI sensor in a feedback loop. It functions as an oscillator whenever a voltage is applied. The period of the circuit's oscillation varies with the inductance of the MI sensing coil and therefore the external field. When no external magnetic field is applied, driving the circuit with a positive (forward) or negative (reverse) voltage will yield the same oscillatory period ($\tau$). However, if there is a field present, the oscillatory period for forward biasing the circuit ($\tau_P$) and reverse biasing the circuit ($\tau_N$) will be different (Figure 2). Measuring the time to complete a cycle in both directions and taking the difference yields a value that can be directly related to the magnetic field.

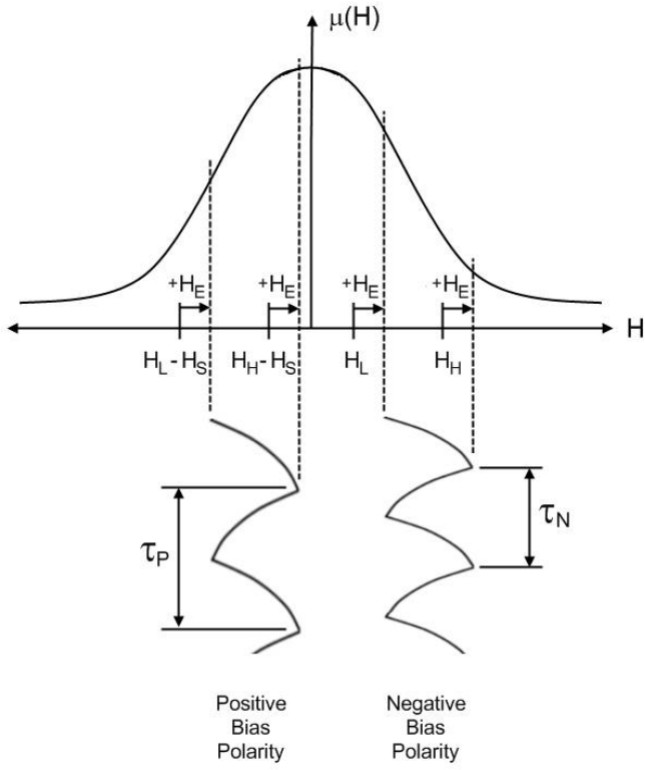

**Figure 2.** MI Sensor Circuit with External Field Present (from Leuzinger and Taylor (2010))

The novel underlying principle of this technology, in which the magnetic field is determined solely by the time difference between forward and reverse biased cycles, provides a completely digital measurement without the use of an analog-to-digital converter (ADC) nor an amplifier. These components are weak-points of traditional magnetometers and their elimination significantly decreases the power budget and failure rate of the instrument. Additionally, the simple oscillatory circuit and components that drive the technology are well-suited for mass production, lowering the cost to produce sensors significantly.

These advantages are key criterion for deployment in future multi-CubeSat missions to potentially study the dynamics of planetary magnetospheres and the solar wind.

## 3   Single Sensor Performance

The performance of a single RM3100 magnetometer was previously extensively studied. Table 1 summarizes the primary char-
acteristics of the sensor as presented by Regoli et al. (2018b). It should be noted that the demonstrated sampling frequency and corresponding resolution of the sensor present applicability to the study of ULF magnetospheric waves in the PC4-PC5 range. With that said, a resolution improvement of at least $2x$ is required for deep-space missions where the magnetic field is on the order of 1-10 $nT$ (Primdahl, 1979) and upwards of $20x$ improvement for the instrument to observe PC1 waves. The area, weight, and power consumption of the instrument alone, however, open the door to CubeSat missions and power-limited
ground-based systems (remotely operated vehicles, planetary landers, or extreme Earth-based environments). The sensor has already been employed in both terrestrial (Shahsavani and Vafaei, 2020) and aeromagnetic (Shahsavani, 2021) geological surveys of iron ore deposits, demonstrating the applicability of the RM3100 to geomagnetic, space physics, or other magnetometer application.

**Table 1.** PNI RM3100 Characterization (adapted from Regoli et al. (2018a))

| Parameter | Value |
|---|---:|
| Dimensions | 2.54 $cm$ x 2.54 $cm$ |
| Mass | < 3 $g$ |
| Power Consumption | < 10 $mW$ |
| Dynamic Range | +/- 100,000 $nT$ |
| Sampling Rate | 40 $Hz$ |
| Resolution @ 40 $Hz$ | 8.73 $nT$ |
| Resolution @ 1 $Hz$ | 2.7 $nT$ |
| Noise Floor | 4 $pT/\sqrt{Hz}$ @ 1 $Hz$ |

Beyond the baseline features presented in Table 1, a surprising but nonetheless valuable additional characteristic of the
RM3100 is its relative radiation hardness. Regoli et al. (2020) irradiated nine separate sensors at two facilities using different dose rates up to a total ionizing does (TID) of at least 300 $krad$ (SI). Of the nine sensors, only two failed during irradiation (the lowest at a TID of 150 $krad$) with one recovering in the month immediately following exposure. It should also be noted that an appreciable difference in resolution was not observed in comparing pre and post irradiation measurements for working magnetometers. Being robust up to 150 $krad$ (SI) enables its use in a variety of space environments including potential missions
to the Jovian moons or Van Allen radiation belts where TID is expected to be high for typical mission lengths (Boudenot, 2007; Regoli et al., 2020). In addition, tests for destructive single-event effect susceptibility of the PNI RM3100 magnetometer sensor

were conducted using the heavy ion beam at the Lawrence Berkeley National Laboratory's 88" Cyclotron. The tests found no single event latchup events for LET > 75 $MeV cm^2/mg$ at an elevated temperature of 85 $^\circ C$ (Moldwin et al., 2022).

Leuzinger and Taylor (2010) make the comment that the output of RM3100 will be inherently stable over temperature due to the forward/reverse biasing nature of the MI circuit. Experimentally, this is not what has been observed. Tests are currently being carried out by the University of Michigan Moldwin Magnetics Lab to fully characterize the gain of the sensor over the temperature range -35 $^\circ C$ to 80 $^\circ C$. Preliminary results show that the thermal gain is significant (roughly 0.5 $nT$ per $^\circ C$) and has some nonlinear behavior. With that said, the non-linearity is repeatable and consistent, enabling its removal through a simple correction.

The fundamental limitation of the RM3100 lies in its inability to detect sub-$nT$ field changes. Low amplitude ULF waves are currently inaccessible as a result, requiring the instrument to detect wave amplitudes on the order of 0.1 $nT$. Efforts at the University of Michigan to improve resolution have fallen under two categories, namely, developing a new instrument based on the MI principle and creating an array of COTS sensors to oversample in space. This paper explores the second solution.

## 4   Quad-Mag Board

The Quad-Mag integrates four independent RM3100 magnetometers (outlined in purple in Figure 3a), commercially developed by PNI, on a single board. The inclusion of four RM3100s permits oversampling with multiple sensors (as opposed to traditional over-sampling in time) and an improvement in the resolution of the instrument by a factor of two ($\sqrt{N}$ improvement, where $N$ is the number of sensors), without sacrificing the sampling frequency necessary to detect ULF waves in the Earth's magnetosphere. The fully functional, flight-ready system presented here utilizes an MSP430FR5949 microcontroller to synchronize sensor data and provide a flexible, streamlined command interface between the host computer and instrument. The MSP430 controller product line was specifically chosen for its Ultra-Lower Power Modes, processing speed, and overall versatility (Strange, 2006; Konte et al., 2018). It has been used in numerous missions (e.g., Li et al., 2020), including the first Mars Cubesat mission where two MSP430's coordinated the Command and Data Handling (CDH) system (Schoolcraft et al., 2016). Controllers in this family have also been irradiated, with the MSP430FR5739 microcontroller surviving up to a TID of nearly 250 $krad$ (Netzer et al., 2014).

The assembled board has dimensions 10 $cm$ x 10 $cm$ x 3 $cm$. The board's mass, including electronics, is 59.05 g. The footprint was designed such that the instrument can fit into one slot of a 1U CubeSat. From Figure 3b, the yellow box highlights the headers for communication with the instrument and the red box outlines the power connector. The current design requires six wired connections to the CubeSat (two UART lines, two debugging lines, and power/ground lines). This could easily be modified to allow the board male connectors to slot into a female connector on the CDH bus of the satellite, eliminating wire connections. A previous board iteration, meant to be flown on the cancelled Michigan Bicentennial Archive CubeSat (M-BARC), with this architecture is displayed in Figure 4. The 90 degree connector required for seamless integration with a CubeSat CDH bus would be soldered to the pads outlined in red.

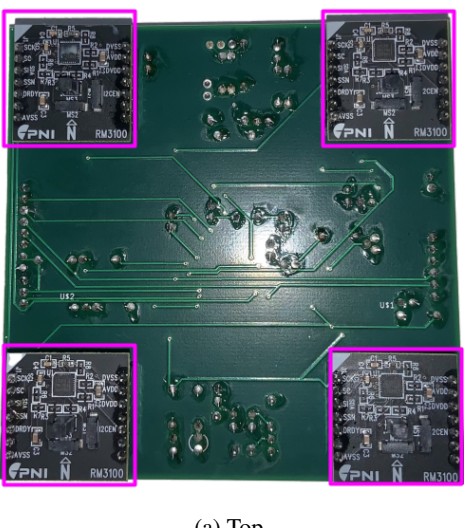
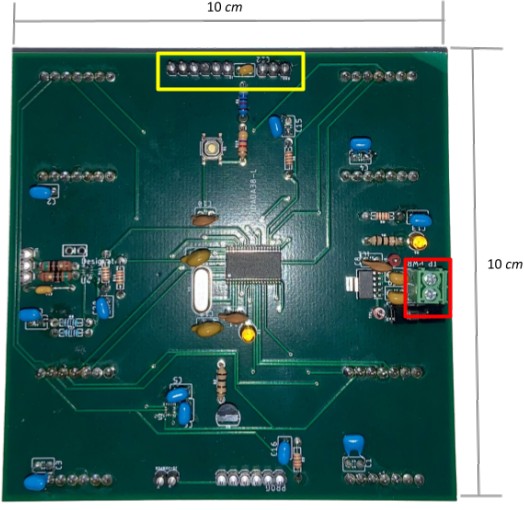

(a) Top                                      (b) Bottom

**Figure 3.** Fully-Assembled Quad-Mag Board (mass of 59.05 $g$)

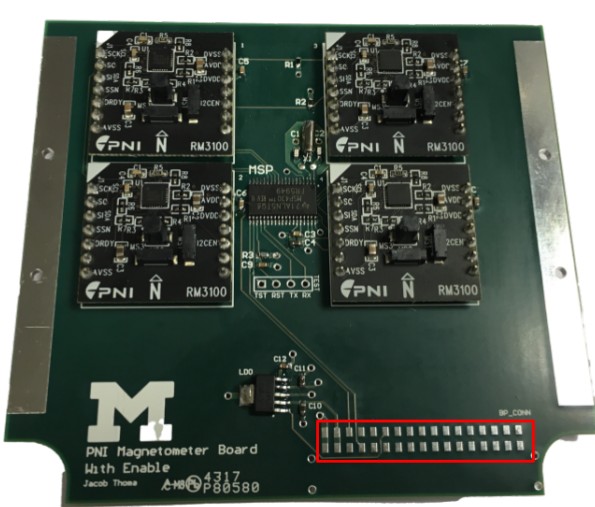

**Figure 4.** Quad-Mag Board for M-BARC CubeSat Mission (adapted from Regoli et al. (2018a))

### 4.1 Operation

Data collection and processing with the instrument is meant to be simple and flexible. Specifically, the powerful microcontroller coordinating the system allows all low-level functionality to be abstracted into a single serial byte stream. This serial byte stream employs the Universal Asynchronous Receiver-Transmitter (UART) communication protocol at a speed of 115.2 $Kbps$. Connection to the instrument is then made by a host computer (for a CubeSat this would be the CDH system) at the other end of the point-to-point UART bus. The versatility of the MSP430 also allows for the serial byte stream to instead use the Serial Peripheral Interface (SPI) protocol or the Inter-Integrated Circuit (I2C) protocol in case of conflicts or need for a higher communication speed. In either case, the system operates in a command/response format. The host has access to a preset list of commands that can be sent to and processed by the instrument. Each command then requires a specific response to be sent to the host in the form of an acknowledgment or data. The outline of the command/response packets are visible in Figure 5. The command header is a unique identifier assigned to each of the commands provided to the host. Similarly the data header allows the host to distinguish between different types of responses provided by the controller. These mappings are known prior to operation and enable both the host and controller to properly parse the data packet following the header. A checksum is included in the response packet of the controller to confirm the integrity of data transmission.

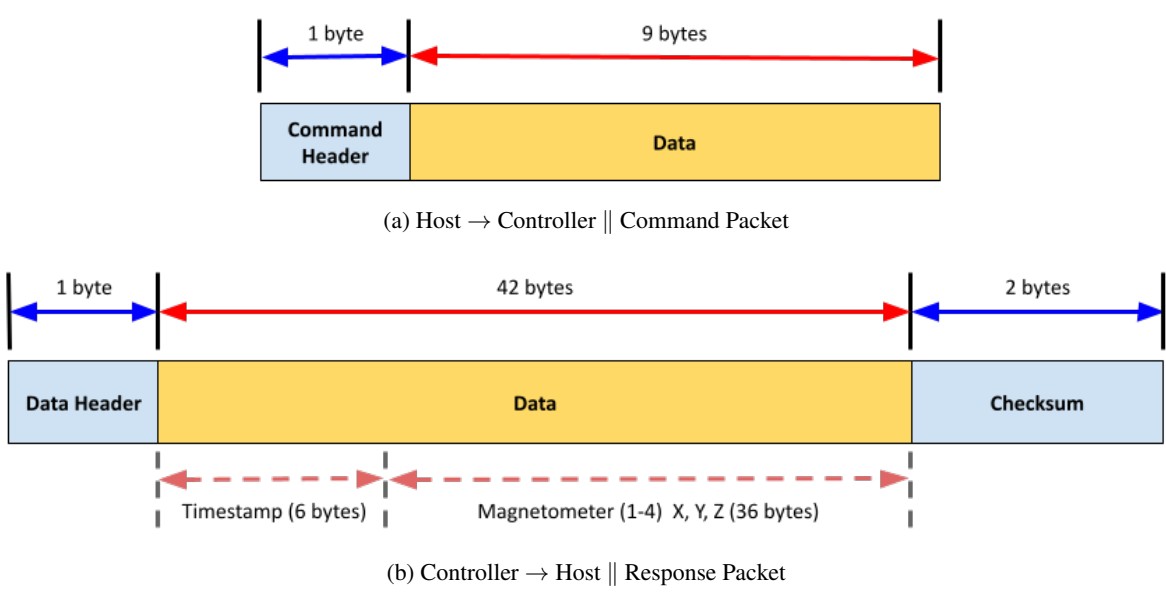

(a) Host → Controller ‖ Command Packet

(b) Controller → Host ‖ Response Packet

**Figure 5.** Packet Format for Communication with the Quad-Mag

The available commands can be placed into two classes, namely, setup and measurement. Setup commands control the various adjustable parameters of the magnetometers, e.g., cycle count and sampling rate, while measurement commands simply retrieve data from each of the sensors. The measurement commands can further be broken down into continuous and single mode in which data can be streamed for a period of time for the former or individual measurements can be requested for the

latter (particularly useful if an atypical sample rate is desired). Of these two modes, continuous has the most common use-case, e.g., if the instrument is functioning as part of an Attitude Determination and Control System (ADCS). Typical operation of the instrument would consist of the host sending setup commands (confirming the command was executed via the instrument's response) and then requesting data through a continuous or single measurement command (which would be followed with a

data packet or stream of data packets as outlined in Figure 5b).

Behind the scenes, each of the four RM3100 magnetometers are attached to one the of available MSP430 SPI buses, with interrupt lines attached to General Purpose In/Out (GPIO) pins of the controller. When operating in either continuous or single measurement mode, the four magnetometers are signaled to take readings simultaneously. The interrupt lines of each sensor are asserted when a measurement is ready. This assertion is received through the GPIO of the MSP430 and used to precisely

timestamp each measurement. The arrival time separation for all four magnetometers is typically less than 500 $\mu s$. After all four sensors have been queried and a measurement stored from each, the controller packages the data into the 45 byte response packet outlined in Figure 5b. This packet-processing takes on the order of 2-3 $ms$. The direct consequence of these delays is periodic skipped/missed readings from all four sensors. This is clearly apparent in the output data rate of the Quad-Mag with it being roughly 15% slower than a lone RM3100 magnetometer possessing identical settings (at frequencies below 100 Hz).

Despite this, the instrument still easily covers the frequency range necessary to study ULF waves in the Earth's magnetosphere, with a maximum observed sampling rate of roughly 222 $Hz$ querying each axis of all four magnetometers.

## 4.2  Characterization

The single RM3100 magnetometer was previously characterized in a controlled laboratory setting using a number of tests including linearity, frequency response, resolution, stability, and radiation (Regoli et al., 2018b, 2020). As stated previously, a

complete thermal characterization of an individual sensor is also currently underway. The focus of this research, however, is on improving the resolution of the instrument and therefore radiation, linearity, and frequency response tests will be omitted in the characterization of the Quad-Mag presented here. An additional interference test that explores the magnetic influence of multiple sensors and board electronics was also carried out.

Theoretically, placing four magnetometers on a single board should allow an overall improvement in the resolution of the

instrument by a factor of two through oversampling with multiple sensors ($\sqrt{N}$ improvement, where $N$ is the number of sensors). However, there is the issue of introducing magnetic noise from the microcontroller and other components necessary to run the system. In order to mitigate this, the board was designed such that the RM3100's are placed as far away from noise producing components as possible, i.e., opposite sides of the board (Figure 3).

The testing presented in the following sections was carried out at the University of Michigan Department of Climate and

Space Science and Engineering (CLaSP). Specifically, the tests involved placing the Quad-Mag inside a three-layer $\mu$-metal lined Zero Gauss Chamber that was in turn placed in a $\mu$-metal-lined-copper room (Figure 6). This is the same setup used by Regoli et al. (2018b) with the only difference being the Quad-Mag uses a sampling rate of 65 Hz as opposed to 40 $Hz$. The higher sampling frequency is a result of using a lower cycle count (400 for all experiments presented here as opposed to 800

previously used). The decision to use this lower cycle count was based on empirically observed lower noise at 1 $Hz$ (post decimation) and the ability to detect higher frequency signals with the resulting increased sampling rate.

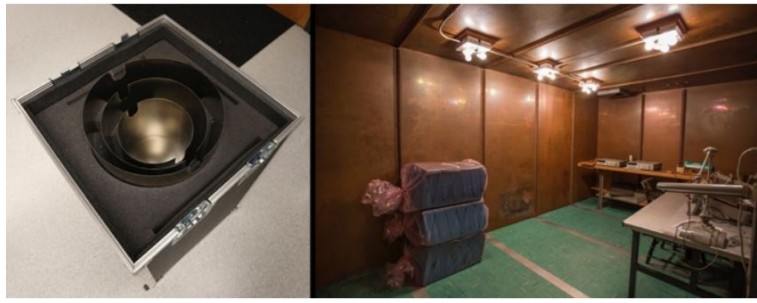

**Figure 6.** Zero Gauss Chamber (left) and Copper Room (right) used for Resolution Test (from Regoli et al. (2018b))

It was mentioned that the magnetic field of an MI sensor is a combination of the internal field (generated by electrical components) and the external field. The internal field presents itself as a unique offset inherent to each magnetometer. This offset was calculated by placing the Quad-Mag inside the Zero Gauss Chamber and taking the difference between the measured field and the residual field determined by a Meda uMAG fluxgate magnetometer (MEDA, 2005). All data sets presented in this paper have the calculated offsets removed. The values of these offsets generally range from a few hundred $nT$ to a few thousand $nT$, depending on the axis. In practice, the internal offset changes slightly after every power cycle of the sensor due to its digital components. As a result, the Quad-Mag board requires careful calibration if used for absolute field measurements.

## 4.3 Resolution

Determining the minimum field that can be detected by the instrument is straightforward. The Quad-Mag board was placed inside the Zero Gauss Chamber in the copper room (Figure 6) such that any remnants of the Earth's magnetic field and other fields generated from nearby current carrying wires (e.g., AC power lines) were sufficiently removed. The resulting time-varying field experienced inside the chamber is well below the noise floor of the instrument.

The system was initially configured to take measurements at a sampling rate of 65 Hz for 30 seconds. In addition, a ten minute warm-up period was undertaken where data were requested from the sensors but not recorded upon being received. This was to allow the system/ambient temperature to settle to a constant value such that changes in gain related to temperature could be avoided as much as possible. The standard deviation of the measured signal is accepted as the resolution of the instrument (minimum signal to be detected) for the given sampling frequency. Figure 7 displays the time series results for this test across all three axes. The plots overlay the measurement taken from each magnetometer and the average of all four measurements (oversampling with all four sensors) in different colors (see legend for details). Additionally, at the top of the plots, the resolution for each individual magnetometer is listed in order, followed by the average of all four resolution values and then the resolution of the combined measurements. The average of all four resolution values is simply an arithmetic mean of the four previously listed standard deviation values. This is compared to the resolution (standard deviation) of the combined

measurements where the data sets of all four magnetometers are stacked (i.e. added together along each axis) and the arithmetic mean taken.

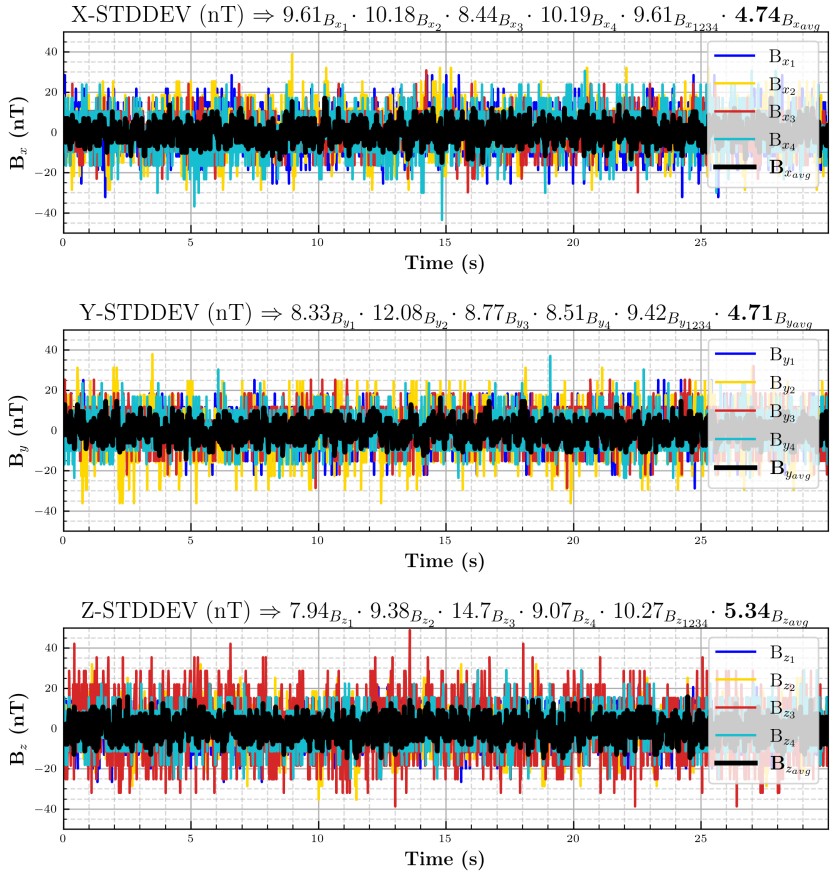

**Figure 7.** Quad-Mag Resolution Test Data @ 65 Hz

There are a few important takeaways from Figure 7. First, the average standard deviation of the axes are relatively close together. There is only a 190 $pT$ difference between the $X$ and $Y$ axes and at most an 850 $pT$ difference between the Z and
5  the other two. Second, when determining the theoretical improvement in resolution for the Quad-Mag, oversampling with four sensors should yield a $2x$ improvement. Comparing the average standard deviation of each axis to the standard deviation of the average of the four measurements, it can be seen that this holds very close to true. The worst improvement in resolution is seen in the $Z$ axis of the Quad-Mag ($1.9x$), while the other two axes both show nearly exactly a $2x$ improvement in resolution. The anomalies seen in the $Z$ axis can be attributed to the relatively large standard deviation of the third sensor (50% higher than
10  the other three sensor standard deviations). The resolution at 65 $Hz$ for the three axes is taken as 4.74 $nT$ ($X$-axis), 4.71 $nT$ ($Y$-axis), and 5.34 $nT$ ($Z$-axis). This is already well below the 6.74 $nT/LSB$ digital resolution of the instrument.

The resolution of the system at 1 $Hz$ was also calculated. To quantify this, a test was run with the Quad-Mag sampling at 65 $Hz$ for ten minutes. The resulting data were then down-sampled to 1 $Hz$ using the typical approach of passing the signal through a low-pass filter and then decimating by an integer factor. As with the previously described test, a ten minute warm-up period was undertaken immediately before data were collected. The results of this test are shown in Figure 8. As with the previous figure, the plots overlay the measurement taken from each magnetometer as well as the average of all four measurements (oversampling). At the top of each plot is the resolution for all four sensors, followed by the average of these four resolutions and then the resolution of the combined measurements.

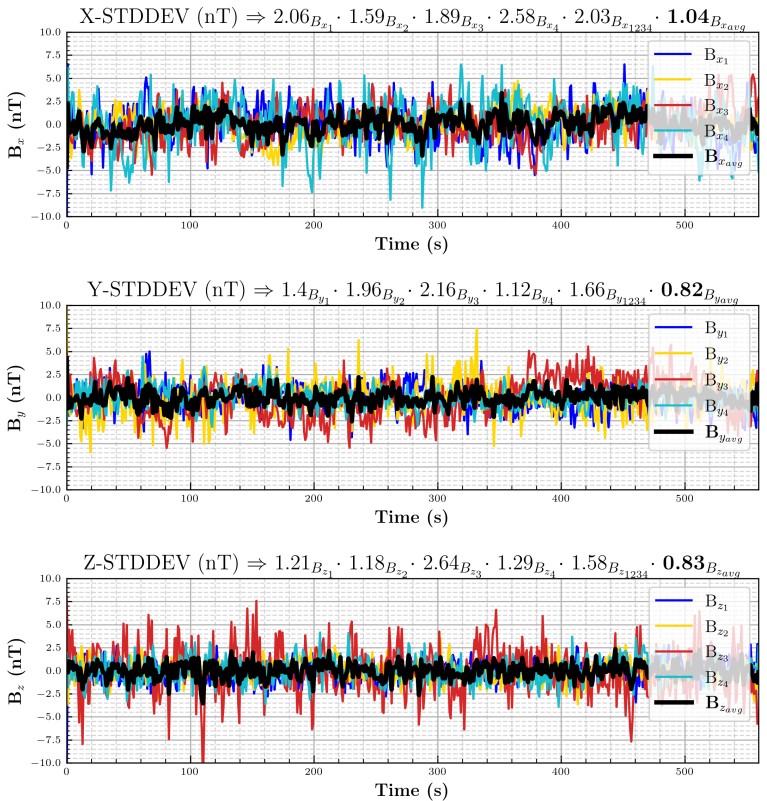

**Figure 8.** Quad-Mag Resolution Test Data @ 1 Hz

Although it can be seen in Figure 8, it should be noted that outliers have not been removed from the data (this is most obvious in the $Z$-axis time series where multiple single point spikes are present). Interestingly, after decimation, the $X$-axis has the largest standard deviation of the three (this can be attributed to the actual decimation process). As before, the resolution improvement can be quantified by comparing the average standard deviation and the standard deviation of averaging all sensors for each axis. The $Z$-axis still shows the largest difference from the theoretical $2x$ improvement, in this case displaying only a

1.9x improvement in resolution over an individual RM3100. The established resolution at this sampling frequency is taken as 1.04 $nT$ ($X$-axis), 0.82 $nT$ ($Y$-axis), and 0.83 $nT$ ($Z$-axis).

In addition to establishing the resolution of the Quad-Mag (standard deviation of the measured signal), it is also valuable to determine the noise floor of the instrument as another performance characteristic. This is calculated from the power spectral density (PSD) of the measured signal inside the Zero Gauss Chamber in the copper room. The formulation of the PSD is not trivial and there are multiple methods that yield different results (Heinzel et al., 2002). For example, Miles et al. (2019) use a unit-correct implementation of Welch's method (Welch, 1967) that yields a value orders of magnitude higher than the method presented in Regoli et al. (2018b).

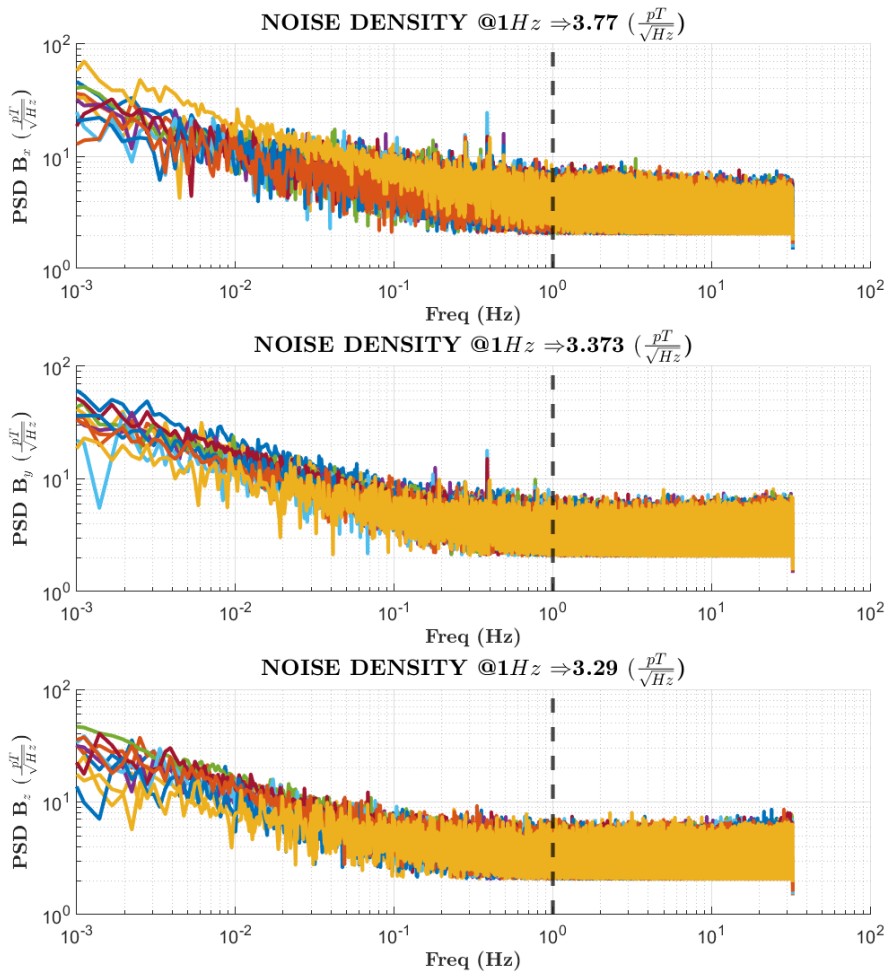

**Figure 9.** Quad-Mag Power Spectral Density

To make a more direct comparison, this paper follows Regoli et al. (2018b) in which the PSD is produced from the Fourier transform of the auto-correlation function of the measured signal. Due to the $1/f$ dependence of the output, the noise floor of the signal is taken as the value of the PSD at 1 $Hz$. The system was again configured to sample at 65 $Hz$, this time for 1 $Hour$. The test was run a total of ten times and the average of the ten runs was used. From this, the noise floor of the Quad-Mag was determined to be 3.770 $pT/\sqrt{Hz}$ ($X$-axis), 3.373 $pT/\sqrt{Hz}$ ($Y$-axis), and 3.290 $pT/\sqrt{Hz}$ ($Z$-axis) as seen in Figure 9.

### 4.4 Stability

As noted previously, the Quad-Mag measurements are not stable between power cycles. The offsets of each sensor axis are different each time the sensor is powered on. This is not an issue in the presented testing as all offsets are removed. In the case of requiring absolute field measurements, however, careful calibration will be needed. Although the offsets are not constant between power cycles, they should be for a single power cycle. This helps define the stability of the sensor which can be described by the variation in output while experiencing an ideal, constant input. For the Quad-Mag, this can be measured under the condition of the instrument experiencing no external field. Placing the Quad-Mag inside the Zero Gauss Chamber, inside the copper room achieves this. The system was configured to take measurements at 65 $Hz$ for roughly 38 $hours$.

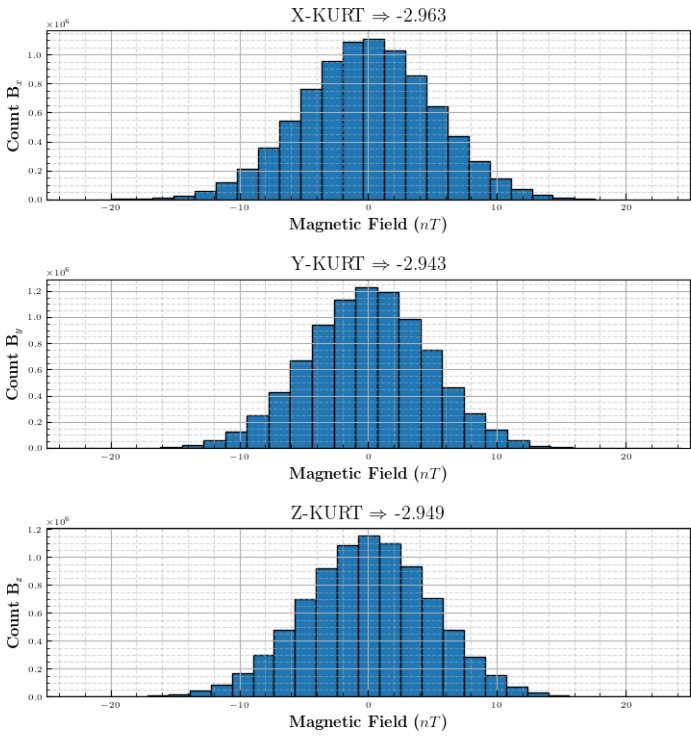

**Figure 10.** Quad-Mag Stability Test

Figure 10 displays the distribution of measurements from this test for each axis. The bin size was set to 1.685 $nT$ (this is not arbitrary but rather equal to LSB / 4). From this figure, the distribution of all axes appear to be Gaussian (as would be expected with white noise). The randomness of the output is confirmed via calculation of the Kurtosis index, which should be close to -3 for a standard normal distribution with light tails (Fiori and Zenga, 2009). In this test, the indices are -2.963 ($X$-axis), -2.943 ($Y$-axis), and -2.949 ($Z$-axis). The Quad-Mag is clearly extremely stable over time.

## 4.5 Interference

The construction of the Quad-Mag introduces two potential magnetic noise sources for each of the four sensors on the board. These two new sources are the companion magnetometers and the board electronics respectively. It is well documented that spacecraft and sensor electronics will present an offset in magnetic readings (Singer et al., 1996). In this case, the offset is removed and thus unimportant. Rather, the effect on resolution must be quantified. To determine this for companion magnetometers, first, a single RM3100 was placed on the quad-mag board inside the Zero Gauss Chamber in the copper room. Measurements were taken for 30 seconds at 65 $Hz$ with a ten minute warm-up period immediately prior. Next, all four magnetometers were placed on the board and similarly configured to sample for 30 seconds at 65 $Hz$ following a ten minute warm-up period. The results of these two tests are shown in Figure 11.

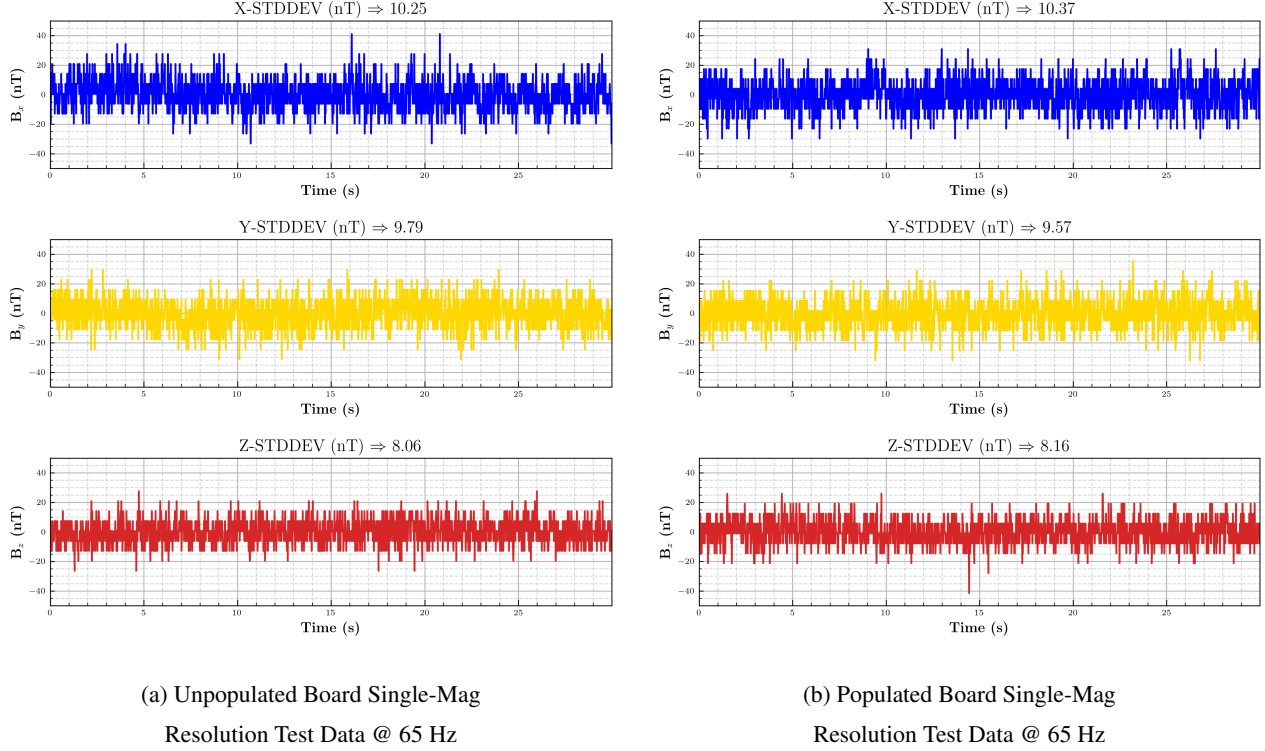

(a) Unpopulated Board Single-Mag
Resolution Test Data @ 65 Hz

(b) Populated Board Single-Mag
Resolution Test Data @ 65 Hz

**Figure 11.** Companion Magnetometer Interference Test

At the top of each plot is the resolution of the specified sensor axis in $nT$. Direct comparison of Figures 11a and 11b shows minute differences in the resolution for each axis. The most pronounced deviation is in the $Y$-axis where we can see a $0.22\,nT$ variation between the two scenarios. This is well below the established resolution of an individual sensor (Regoli et al., 2018b) and clearly implies that multiple magnetometers on the same board do not have significant influence on each other's resolution.

The second potential noise source stems from the controlling electronics on the board, most notably the MSP430 microcontroller. To understand the board's effect on the resolution of the instrument, it is sufficient to look at its effect on a single magnetometer. This is determined by taking the same magnetometer used for the previous test and placing it inside the Zero Gauss Chamber in the copper room without the Quad-Mag present. Measurements were then taken for 30 seconds, with identical configuration settings as in the prior test. Due to the processing delay introduced by the Quad-Mag, the sampling rate for

these measurements is about 15% higher at $78\,Hz$. A ten minute warm-up period was again undertaken. The results of this test can be seen in Figure 12b. Figure 12a is copied from Figure 11a for ease of comparison. At the top of each plot is the resolution of the specified sensor axis in $nT$.

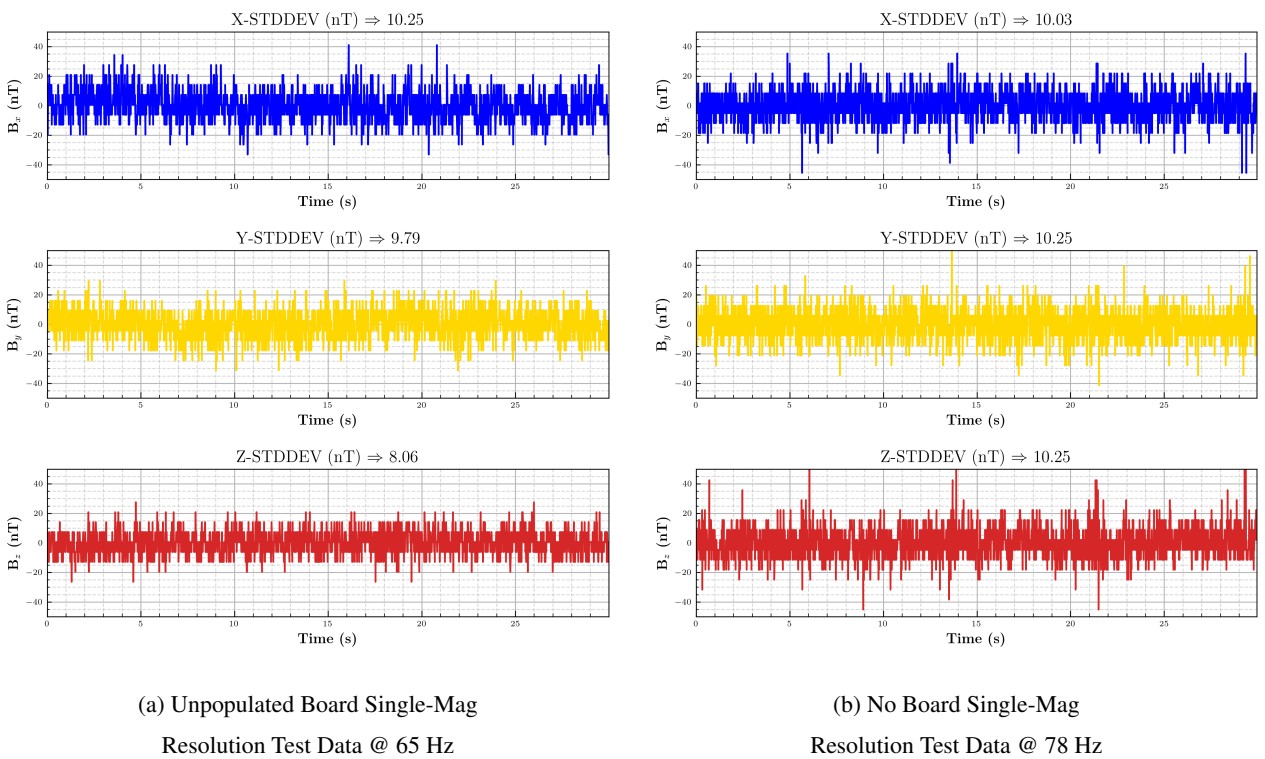

(a) Unpopulated Board Single-Mag
Resolution Test Data @ 65 Hz

(b) No Board Single-Mag
Resolution Test Data @ 78 Hz

**Figure 12.** Board Interference Test

The results shown in Figure 12 exhibit a $0.22\,nT$ ($X$-axis), $0.46\,nT$ ($Y$-axis), and $2.19\,nT$ ($Z$-axis) difference in resolution compared to the previous testing setup. Although the largest difference here is two orders of magnitude higher than what is

seen in Figure 11, it is still well below the established resolution of a single magnetometer at these sampling frequencies and

thus demonstrates a lack of significant interference generated by the board electronics. It should also be highlighted that the sampling frequencies in this experiment are not identical due to processing delay introduced by the quad-mag. As such, the larger differences in resolution could be explained partly by this disparity.

## 5   Discussion

The characteristics of the Quad-Mag are in Table 2. A standalone RM3100 has a resolution of around 2.2 $nT$ @ 1 $Hz$. The Quad-Mag board presented reduced that to 1.04 $nT$ or less for each axis through oversampling with four sensors. This actually exceeds the theoretical 2x improvement expected by about 10%. A closer examination of the resolution of each axis on each magnetometer reveals that there is a large disparity between the resolution of individual sensor coils. In fact, the resolution of individual coils was found to vary between 1.12 $nT$ and 2.64 $nT$ @ 1 $Hz$. This disparity most likely arises from variations

in the met-glass material of the sensing coil that naturally occur due to the manufacturing process. The consequence of this is that certain sensors perform better or worse than others. Drawing from this conclusion, the extra 10% improvement in resolution was most likely a result of using inherently better RM3100s than used in the initial characterization of Regoli et al. (2018b). The established resolution (Table 2) is therefore taken as an upper limit. Depending on the specific requirements of the measurements, it is recommended that individual RM3100's are characterized in batches of 20 to find the the lowest noise

and highest resolution sensors that achieve the best measurements.

**Table 2.** Quad-Mag Characteristics

| Parameter | Value |
|---|---:|
| Dimensions | 10 $cm$ x 10 $cm$ x 3 $cm$ |
| Mass | 59.05 $g$ |
| Power Consumption - Sampling | 23 $mW$ |
| Power Consumption - Inactive | 14 $mW$ |
| Dynamic Range | +/- 100,000 $nT$ |
| Sampling Rate | 65 $Hz$ |
| Resolution @ 65 $Hz$ (Individual Axis) | 5.34 $nT$ |
| Resolution @ 1 $Hz$ (Individual Axis) | 1.04 $nT$ |
| Noise Floor (Individual Axis) | 3.77 $pT$ / $\sqrt{Hz}$ @ 1 $Hz$ |

## 6   Conclusions

A Quad-Mag board containing four PNI RM3100's and an MSP430 microcontroller is presented. The primary purpose of the board is the detection and study of the Earth's geomagnetic field, field-aligned currents, and ULF waves. The board particularly excels in its low size, cost, and power savings over other magnetometer options. These qualities make the board naturally well

suited to CubeSats, for which size, power, and cost are at a premium. A simple UART interface is provided for the CubeSat to both send commands to the board and read data. The Quad-Mag is also an option for ground-magnetometers at high-latitudes where 1 $nT$ resolution @ 1 $Hz$ is sufficient to measure most geomagnetic disturbances of interest.

The quad-board could theoretically be extended to any number of magnetometers, depending on the requirements of the mission. It would likely be cheaper in both cost and weight to include a large array of RM3100 magnetometers to improve instrument resolution than to include a typical flux-gate. In combination with the development of a new, higher resolution MI magnetometer, the multi-magnetometer array strategy appears to be a viable low-cost and power option for the study of ULF waves, particularly on small satellites or remote, power-constrained ground based systems.

The current design of the system requires it to be placed inside a CubeSat and thus forces it to be subjected to noise from the spacecraft. Different approaches have and are being undertaken to solve this issue. A promising solution that has recently been investigated involves using an underdetermined blind source separation (UBSS) algorithm to identify and remove noise generated by various unknown sources (Hoffmann and Moldwin, 2022).

*Code and data availability.* The data produced during the characterization of the instrument can be downloaded from https://doi.org/10.7302/kc6t-3670 (Strabel, 2022b). The code used to run the system and analyze the data is available at https://doi.org/10.5281/zenodo.6515198 (Strabel, 2022a).

*Author contributions.* Brady P. Strabel, Mark B. Moldwin, and Lauro V. Ojeda designed the experiments and Brady P. Strabel executed them. Brady P. Strabel, Lauro V. Ojeda, and Leonardo H. Regoli developed the code for and carried out the data analysis. The manuscript was prepared by Brady P. Strabel with contributions from all co-authors. Isaac S. Narrett, Jacob D. Thoma, and Matthew Pellioni were students at the University of Michigan at the time of their contribution to this research.

*Competing interests.* The authors declare that they have no conflict of interest.

*Acknowledgements.* This works was partially supported by NASA grants 80NSSC18K1240, 80NSSC19K0608 and NSF AGS 1848724.

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
