# Peer review of "Quad-Mag Board for CubeSat Applications"

_EGUsphere, 2022_

## Author Response (AR1)

* * *
Editor's Initial Comments -> 05/09/2022
* * *
Really, this is a very interesting paper that will be interesting for students, engineers, and scientists. Nevertheless, I want to clarify a few questions.
(1) I propose that a detailed explanation must be given of the fact, which preferences we received from the construction of the Quad-Mag Board.
(2) Can the ULF measurements potentially help to register earthquake precursors?
(3) The Authors speak about the space and possibly ground magnetometers application. And what is it about Remote Operated Vehicles?
(4) List of Literature for such a paper must be extended.
* * *
Author's Responses to Initial Comments -> 05/12/2022
* * *
(1) Assuming this is asking about "lessons" learned or perhaps "advances" made by construction of the sensor, after line 10 page 2 we added, "The system presented in this paper takes the latter approach and describes not only the use of a low-size, weight, and power plus cost (SWAP+C) chip-based COTS magnetometer, but the combination of multiple magnetometers into a single system to improve the resolution by oversampling with multiple sensors."

(2) There is no compelling evidence of magnetic precursors to Earthquakes (e.g., Can Wang, Lilianna E Christman, Simon L Klemperer, Jonathan M Glen, Darcy K McPhee, Bin Chen, Assessment of a claimed ultra-low frequency electromagnetic (ULFEM) earthquake precursor, /Geophysical Journal International/, Volume 229, Issue 3, June 2022, Pages 2081–2095, https://doi.org/10.1093/gji/ggab530)

(3) The quad mag could be used in any application where a conventional magnetometer can be used including vehicles. Added a section starting at line 6 on page 4 ". With that said, a resolution improvement of at least 2x is required for deep-space missions where the magnetic field is on the order of 1-10 nT  (Primdahl, 1979) and upwards of 20x improvement for the instrument to observe PC1 waves. The area, weight, and power consumption of the instrument alone, however, open the door to CubeSat missions and power-limited ground-based systems (remotely operated vehicles, planetary landers, or extreme Earth-based environments). The sensor has already been employed in both terrestrial  (Shahsavani and Vafaei, 2020) and aeromagnetic (Shahsavani, 20) geological surveys of iron ore deposits. The RM3100 excels in any geomagnetic, space physics, or other magnetometer application."

(4) Additional references added to line 19 page 1, line 10 page 2, line 8 and 11 page 4, and line 19 page 5.
* * *
Reviewer #1 Comments -> 05/26/2022
* * *
Manuscript describes characteristic and performance of a magnetometer board (Quad-Mag) equipped with four PNI RM3100 magnetometers.
Description is detailed and clear.
To my mind, there are two technical characteristics that should be given consideration:
a) Mutual influence of magnetic sensors placed on the same board (could be evaluated, for instance, with the help of another magnetometer board)
b) Magnetic influence (interference) of board electronics on magnetic measurements being taken (probably similar to a))
Technical:
*page 5, line 4 - probably, term "thermal drift" instead of "thermal gain" is more customary? (and further in the text);
*page 5, line 12 - should be Figure 3a
*page 13, line 1 - should be "configured to gain sample"
* * *
Author's Responses to Reviewer #1 Comments -> 07/07/2022
* * *
a) Added section "Interference" to address concerns of mutual influence of magnetometers and of board electronics on measurements.
b) See above
Technical:
*page 5, line 12 - changed to figure 3a
*page 13, line 1 - changed to "the system was again configured to sample"
* * *
Additional Author Changes -> 07/25/2022
* * *
*Changed number of sig figs to represent resolution from 3 to 2
*Updated text as well as figures 7, 8, and table 2 to match sig fig change
* * *
Reviewer #2 Comments -> 09/29/2022
* * *
General comments: An inexpensive and lightweight magnetometer concept is presented. Losses in quality compared to conventional instruments are explained in the preprint. The quality of the instrument is examined with comprehensible tests.

A very similar article was presented in 2018 as "Investigation of a low-cost magneto-inductive agnetometer for space science applications" at https://doi.org/10.5194/gi-7-129-2018. What is new here is that four sensors are operated simultaneously.

Stacking the four simultaneous measurements results in a halving of the expected data errors for statistical reasons. The four measurements at different locations on the circuit board would make it possible to identify interference fields generated by the device itself. (As differences in the measurements). The preprint does not go into this direction. But it is mentioned as an outlook ("undetermined blind source separation").

All in all his preprint is relevant and of interest for the community.

Specific comments:

chapter2: principles of magneto inductive sensing is explained. I found it hard to understand. The terms "driving the circuit with a positive (forward) or negative (reverse) voltage ..." puzzled me first. Reversing the supply voltage of a Schmitt trigger is in practice certainly not possible. Looking also at the PNI release notes of the PNI-11096 circuit I understand: Coil, resistor and Schmidt trigger form an oscillator, as properly explained in the preview. During this oscillation, the magnetic core material is driven into saturation. This reduces inductance and accordingly influences the oscillation period. The effect is advanced if the surrounding field is parallel to the field produced by the current in the coil. It is reduced if thefield is antiparallel. The coil is reversed by means of electronic switches in the ASIC included in the RM 3100 and both periods are compared to produce readings of the field.

Chapter 4.2: Is it clear that the offset you measure between different sensors, are due to board-borne constant fields? Later on you state large sensor offsets. Can the sensors be swopped on the board to see if it are the sensors itself?

Chapter 4.4 Stability: You state in chapter 4.2: „The values of these offsets generally range from a few hundred nT to a few thousand nT, depending on the axis. In practice, the internal offset changes slightly after every power cycle of the sensor due to its digital components. As a result, the Quad-Mag board requires careful calibration if used for absolute field measurements." I think this is a stability item an should be mentioned there.

Technical corrections:

Introduction P1L15: The SWARM satellite mission uses three spacecraft to overcome just this problem.

P9L23: please better use the term "stacked" than "overlapped"

In the Abstract the last sentence says:  The Quad-Mag enables 1 nT magnetic field measurements at 1 Hz using commercial-off-the-shelf sensors for space applications. Don't you think this is a bold statement seeing offsets of tenth to hundreds of nT at a single sensor and a resolution above 1nT in a zero Gauß chamber? How about „ The Quad.Mag allows for almost 1nT resolution under optimal conditions."
* * *
Author's Responses to Reviewer #2 Comments -> 10/14/2022
* * *
Chapter2: As outlined by Luezinger and Taylor (2010), the described circuit acts as a comparator-based L/R relaxation oscillator that alternates positive and negative current through the non-grounded side of the sensing coil. Indeed, the reverse and forward biasing is done via electronic switches in the RM3100 control ASIC that effectively change the polarity of the coil. Measurements are taken in both directions and compared to generate a zero-centered, positive/negative field value.

Chapter 4.2: An interference section has been added to address related concerns. In short, the offsets present are in fact a combination of board electronics and mutual sensors present on the board.

Chapter 4.4: Added a paragraph to the Stability section reacknowledging this.

Introduction P1L15: Added an additional reference to the SWARM mission in following paragraph covering recent multi-spacecraft missions.

P9L23: Changed to "stack"

Abstract: Changed to "The Quad-Mag enables nearly 1 nT magnetic field measurements at 1 Hz using commercial-off-the-shelf sensors for space applications under optimal conditions.
* * *
Final Author Changes -> 10/21/2022
* * *
*Added to introduction P2 highlighting how this work expands on previous research
*Citation for newly published UBSS noise cancellation added to conclusion P17
*P1 changed wording slightly for theoretically established limit
*P3 replaced "low resolution" with resolution to remove ambiguity and softened wording of RM3100 applications